# Evolution of a Multiple Sex-Chromosome System by Three-Sequential Translocations among Potential Sex-Chromosomes in the Taiwanese Frog *Odorrana swinhoana*

**DOI:** 10.3390/cells10030661

**Published:** 2021-03-16

**Authors:** Ikuo Miura, Foyez Shams, Si-Min Lin, Marcelo de Bello Cioffi, Thomas Liehr, Ahmed Al-Rikabi, Chiao Kuwana, Kornsorn Srikulnath, Yuya Higaki, Tariq Ezaz

**Affiliations:** 1Amphibian Research Center, Hiroshima University, 1-3-1 Kagamiyama, Higashi-Hiroshima 739-8526, Japan; lizard.dna@gmail.com (S.-M.L.); kornsorn.s@ku.ac.th (K.S.); Tariq.Ezaz@canberra.edu.au (T.E.); 2Center for Conservation Ecology and Genomics, University of Canberra, Canberra, ACT 2601, Australia; Foyez.shams@canberra.edu.au; 3School of Life Sciences, National Taiwan Normal University, No. 88, Sec. 4, Tingzhou Road, Tapei 116, Taiwan; 4Departamento de Genética e Evolução, Universidade Federal de São Carlos, São Carlos 13565-090, SP, Brazil; mbcioffi@ufscar.br; 5Institute of Human Genetics, University Hospital Jena, Am Klinikum 1, 07747 Jena, Germany; Thomas.Liehr@med.uni-jena.de (T.L.); ahmedgenetic@hotmail.com (A.A.-R.); 6Graduate School of Integrated Sciences for Life, Hiroshima University, 1-4-4 Kagamiyama, Higashi-Hiroshima 739-8528, Japan; ahorn27g10@gmail.com (C.K.); yhigaki1320@gmail.com (Y.H.); 7Department of Genetics, Faculty of Science, Kasetsart University, 50 Ngam Wong Wan, Lat Yao, Chatuchak, Bangkok 10900, Thailand

**Keywords:** fusion, autosome, hexavalent, sex-chromosome turnover

## Abstract

Translocation between sex-chromosomes and autosomes generates multiple sex-chromosome systems. It happens unexpectedly, and therefore, the evolutionary meaning is not clear. The current study shows a multiple sex chromosome system comprising three different chromosome pairs in a Taiwanese brown frog (*Odorrana swinhoana*). The male-specific three translocations created a system of six sex-chromosomes, ♂X_1_Y_1_X_2_Y_2_X_3_Y_3_-♀X_1_X_1_X_2_X_2_X_3_X_3_. It is unique in that the translocations occurred among three out of the six members of potential sex-determining chromosomes, which are known to be involved in sex-chromosome turnover in frogs, and the two out of three include orthologs of the sex-determining genes in mammals, birds and fishes. This rare case suggests sex-specific, nonrandom translocations and thus provides a new viewpoint for the evolutionary meaning of the multiple sex chromosome system.

## 1. Introduction

Sex chromosomes generally evolve from an ordinary autosomal pair after acquiring a sex-determining gene and thus are composed of a pair of X and Y chromosomes in an XX-XY system or of Z and W chromosomes in a ZZ-ZW system. Rarely, the sex-chromosome is fused with an autosome and generates multiple sex-chromosome systems. If either a homolog of a sex-chromosome pair is fused with an autosome, the number of sex-chromosomes increases, while if they are both homologs, the pair of sex-chromosomes remains the same but gets larger in size. The latter is the case in placental mammals, including humans, in which an autosome corresponding to kangaroo (marsupial) chromosome 5 was fused with both the X and Y chromosomes [1,2]. To date, the evolutionary meaning of the translocation between sex chromosomes and autosomes has been documented in relation to speciation [3,4], sexual benefit [5], and life elongation of decaying Y chromosomes [6,7]. However, such evolutionary advantages of multiple sex-chromosomes were acquired after the unexpected translocation and thus mean no evolutionary inevitability; that is, evolution has no foresight but always hindsight [8].

In amphibians, a case of multiple sex-chromosomes is very rare. Generally, the karyotypes are highly conserved, with little rearrangement among species [9,10,11]. Likewise, the sex chromosomes are homomorphic in both sexes in the majority of the species [12]. To date, approximately ten cases of a multiple sex chromosome have been reported, all of which show the fusion of homomorphic sex chromosomes with an autosome [13,14,15,16,17,18,19,20]. The unique case is that both systems of the single and multiple-sex chromosomes coexist within a species or even within the same population, for example, in South and Central American frog species [15,16,17]. Although the translocation between homomorphic sex chromosomes and autosomes may be an easy way to quickly achieve heteromorphy of sex chromosomes in amphibians, the evolutionary meaning of building up multiple sex-chromosome systems remains unsolved, as in other eukaryotes.

In 1980, Kuramaoto [13] discovered a male-specific translocation between the two chromosomes, No. 1 and No. 9, in the 13 haploid complements (2n = 26) of the Taiwanese brown frog, *Rana narina* (a synonym for *Odorrana swinhoana*). This is the first report of multiple sex chromosomes in amphibians, and the sex chromosomes can be described as ♂X_1_Y_1_X_2_Y_2_-♀X_1_X_1_X_2_X_2_. This finding suggests that the translocation occurred between the two members of the potential sex-determining chromosomes. The sex chromosomes in amphibians rapidly repeat turnovers during their speciation or the differentiation of geographic populations within a species. The turnover did not occur at random but nonrandomly among six members of potential sex-determining chromosomes, which are No. 1–4, 7, and 9 out of 13 haploid complements [21,22]. Therefore, chromosomes 1 and 9 involved in the translocation in the Taiwanese frog may be members of the potential sex chromosomes. Unfortunately, the identification of the chromosomes involved in the translocation was uncertain because the chromosomes were stained with only a single Giemsa solution, and the identification was based on their size and shape.

In this study, to confirm the male-specific translocation and precisely identify the chromosomes involved in the translocation, we re-investigated the somatic chromosomes as well as meiotic chromosomes of the Taiwanese frog species using chromosome banding and molecular mapping techniques. Unexpectedly, the translocation was found not to be a single but a triple one, comprising potential sex-chromosomes that include orthologs of the sex-determining genes in mammals, birds and fishes.

## 2. Materials and Methods

### 2.1. Frogs

The number of frogs used for this study and the collection locations is shown in Appendix A. One female was mated with a male at the facility and spawned eggs (Taiwan National Normal University). Ten embryos were carried to Hiroshima University and reared until sexual maturation. The sex of the specimens was determined based on the external morphology, such as the black thumb callus of males and eggs of females. Animal care and experimental procedures were conducted with the approval of the Committee for Ethics in Animal Experimentation at Hiroshima University (permit number: G18-2), and Institutional Animal Care and Use Committee (IACUC), National Taiwan Normal University (license No. 107016).

### 2.2. Chromosome Preparation and Banding Techniques

Mitotic and meiotic metaphase chromosomes were prepared from blood cell culture and directly from cells of testes, respectively [10,23]. Late-replication banding and C-banding techniques were basically as previously described, respectively [24,25].

### 2.3. Microdissection and Chromosome Painting Probes Preparation

Fifteen copies of each of the chromosome pairs 1 and 3 from female mitotic cells and hexavalent from male meiotic cells were isolated via manual microdissection and amplified following [26,27,28]. The whole chromosome-derived probes were labeled with spectrum-orange-dutp or spectrum green-dUTP (Vysis, Downers Grove, IL, USA) through 30 cycles of secondary DOP PCR, using 1 μL of the primary amplification product as a template DNA, resulting in a final volume of 20 μL [29]. The final probe cocktail for each slide was composed of 500 ng of the probes and 60 µg of Cot-1 DNA isolated from *O. swinhoana* total genomic DNA (for details, see [30]) to outcompete the hybridization of highly repeated DNA sequences. The hybridization procedure followed our previous studies [30] and was performed for 24 h at 37 °C in a dark and moist chamber. After washing procedures, the chromosomes were counterstained with DAPI (1.2 µg/mL) and mounted in antifade solution (Vector).

### 2.4. Telomere Mapping

Previously described methods [31,32] were followed to determine the chromosomal locations of telomeric (TTAGGG)n sequences. We used biotin-16-UTP labeled TTAGGG repeats as the prove. For hybridization of biotin-labeled probes to *Odorrana swinhoana* chromosomes, slides were first incubated at 65 °C for 2 h in 70% formamide and were rinsed in 70% ethanol for 5 min, in 100% ethanol for 5 min and dried. Then, 10 μL hybridization solution containing 50% formamide, 10% dextran sulfate, 2 × SSC, 40 mM sodium phosphate and 1× Denhardt’s solution were dropped onto the slides with 10 μL probes under a coverslip. After the slides were incubated at 37 °C overnight, they were washed in 50% formamide for 20 min and rinsed in 2 × SSC for 15 min, in 1 × SSC for 15 min, and in 4 × SSC for 5 min. After completed the wash and rinse, the slides were stained with avidin labeled with fluorescein isothiocyanate (Avidin-FITC; Invitrogen, Carlsbad, CA, USA). After incubation at 37 °C for 1 h, slides were rinsed in 4 × SSC and washed on a shaker for 10 min, then in 0.1% Nomident/4 × SSC (NP-40) for 10 min, in 4 × SSC for 5 min and finally in 2 × SSC for 5 min. Slides were subsequently stained with 1 μg/mL 4,6-diamidino-2phenylindole (DAPI). Fluorescence hybridization signals were captured using a cooled Charge-Coupled Device (CCD) camera mounted on a ZEISS microscope (Zeiss, Germany) and processed using MetaSystems ISIS v.5.0. software (MetaSystems, Alltlussheim, Germany).

### 2.5. Comparative Genomic Hybridization (CGH)

CGH was performed to detect genomic sequence differences between males and females. An amount of 500 ng of male and 500 ng of female genomic DNA were labeled with biotin-dUTP and Cy3 dUTP, respectively, using a Nick translation kit (Merck KGaA, Darmstadt, Germany) by incubating at 15 °C for two hours and heating at 65 °C for 10 min. After ethanol precipitation, the slides were denatured by heating at 75 °C for 10 min before 10 μL hybridization solution containing 50% formamide, 10% Dextran sulfate, 2 × SSC, 40 mM sodium phosphate and 1x Denhardt’s solution was dropped onto slides with 10 μL probes. Slides were incubated at 37 °C for 3–4 days in a humid box. Hybridized slides were washed in 4 × SSC for 7 min, in NP-40/4 × SSC for 7 min, and in 4 × SSC for 7 min, and by shaking in 2 × SSC for 5 min. Slides were subsequently stained with 1 μg/mL 4,6-diamidino-2phenylindole (DAPI).

## 3. Results

### 3.1. Three Heteromorphic Sex Chromosomes in Males

To confirm the results of the previous studies [13,14], we investigated karyotypes of 4 males and 3 females collected from the northern population (New Taipei), Taiwan, using late replication banding [25] (Appendix A). All the females showed homomorphic pairs of 13 haploid complements (2n = 26), while all the males had three pairs of heteromorphic chromosomes, which were Nos. 1, 3, and 7 (Figure 1a,b). In the male karyotype, a large part of the long arm of chromosome 1 was missing, whereas the long arm of chromosome 7 was much longer than the homolog. In addition, the short arm of chromosome 3 was slightly longer than the homolog (Figure 1 and Appendix A). We confirmed that the three heteromorphic chromosomes were transmitted to the male offspring in one brood of their F1 generation (Appendix A). These heteromorphic chromosomes suggest that more than one translocation occurred, not conforming to the results of Kuramoto [13,14].

### 3.2. A Hexavalent Ring at Male Meiosis

To investigate the number of translocations, we observed paring figures of the first meiotic chromosomes in male testes. It was found that the meiotic karyotype comprised one large ring-shaped hexavalent, not a tetravalent, together with ten ring-shaped bivalents at the first meiotic metaphase (Figure 1c and Appendix A). This proves that the three chromosome pairs were involved in the translocations and triangularly made a ring, but not an open chain. In addition, we detected telomeres in all the chromosomes involved in the translocations (Appendix A). Therefore, the most likely, expected order of translocations to form the hexavalent ring is as follows: the large part of the long arm of chromosome 1, including the terminal tip, was translocated to the long arm of chromosome 7, in which the terminal tip (including two bands) of the long arm was broken before the fusion and moved to the short arm of chromosome 3, of which the terminal tip (one band) of the short arm was broken before the fusion and moved to the long arm of chromosome 1 that missed the large part of the long arm (Figure 1d). In the remaining parts of the chromosome 1 long arm and of the chromosome 3 short arm, one paracentric inversion occurred (Figure 1d and Appendix A).

### 3.3. Direct Proof of the Chromosome Members Involved in Three Translocations

To directly identify the chromosome members involved in the three translocations, we performed hybridization painting using the microdissected chromosomal DNA probes. Both of the chromosomes 1 and 3 probes (green and red, respectively) painted the parts of hexavalent (Figure 2a(1)). Moreover, then, the chromosome 1 probe painted chromosome pair 1 in female (Figure 2a(2)), and heteromorphic chromosome pair 1 and the long arm of the longer chromosome 7 in male (green arrow in Figure 2a(3)). The chromosome 3 probe painted chromosome pair 3 in female (Figure 2a(2)), and heteromorphic chromosome pair 3 and the short arm of the shorter chromosome 1 in male (pink arrow in Figure 2a(3)). These results directly prove that the two chromosomes 1 and 3 were involved in the male-specific translocations from chromosome 1 to 7 and from chromosome 3 to 1. Next, hybridization using the hexavalent DNA probe painted chromosome pairs 1, 3 and 7 in female, and heteromorphic chromosome pairs 1, 3 and 7 in male (Figure 2a4–a(6)). Thus, it is concluded that the male-specific three translocations among chromosomes 1, 3 and 7 created a system of six sex-chromosomes, ♂X_1_Y_1_X_2_Y_2_X_3_Y_3_-♀X_1_X_1_X_2_X_2_X_3_X_3_ (Figure 2b,c).

### 3.4. No Male DNA Specialization on the Three Y Chromosomes

To examine the extent of chromosomal specialization of the three Y chromosomes, we observed C-banded karyotypes. No Y-specific heterochromatin was detected on the three Y chromosomes (Appendix A). In addition, complementary genomic hybridization (CGH) on the male and female chromosomes detected neither male nor female-specific signals (Appendix A). These results show that the three Y chromosomes have not yet accumulated male-specific DNA or heterochromatin.

### 3.5. Population Variation in the Sex-Chromosome System

To elucidate the sex-chromosomal variations within this species, we observed the karyotypes of the frogs collected from two other populations (Appendix A), of which one was located just south of the first population (New Taipei), and the other was around the central region of the island (Nantou Ren Ai). We observed no translocations or heteromorphic sex-chromosomes in the males or females based on the late replication and C-banding patterns (Appendix A). This result suggests that the three translocations in males are restricted in some geographic populations of the species and originated very recently.

## 4. Discussion

Multiple sex-chromosome systems, which originate from a fusion of sex-chromosomes with autosomes, were discussed from an evolutionary viewpoint. First, if beneficial genes to either sex are located on the autosome involved in the fusion, the genes are changed to a sex-linked state and genetically favor either sex [5,33]. Second, speciation is accelerated; for example, if genes controlling male reproductive behavior are located on the autosome involved in the fusion with the Y chromosome, the male-specific behavior is genetically strengthened by being sex-linked, which enlarges the behavioral differences from males of the original populations bearing no fusion and this accelerates reproductive isolation [3]. Third, fusion with autosomes contributes to the elongation of the life of Y or W chromosomes. By fusion of autosomes to Y or W chromosomes, the size and gene content increase, and thus the life of Y or W chromosomes suffering from genetic degeneration is extended [6,7]. The above advantages might have been materially acquired but are just products from the by-chance fusion incident and thus imply no evolutionary inevitability.

The three sequential translocations specific to males in the Taiwanese frog suggest a different viewpoint from the above to explain the evolutionary meaning, not advantage, of a multiple-sex chromosome system. A unique finding is that the three chromosomes involved in the translocations are all members of the potential sex-chromosomes (Figure 2b–d), which are used for sex-chromosome turnover in frogs [21,34]. The karyotypes of 2n = 26 in true frogs are highly conserved with little chromosome rearrangements [10,21,34]. In fact, the 13 chromosome complements are almost perfectly conserved between *G. rugosa* and *O. swinhoana* based on the late replication banding pattern (Figure 2d). Therefore, each of the three Y chromosomes may include at least one candidate gene for sex determination: these genes are *Dmrt1*, the male determining gene in birds [35], and *Amh*, the male determining gene or candidate in fish and platypus [36,37], on Y1 chromosome (chromosome 1 in *G. rugosa*) [38,39], *Sox3*, the ancestral gene of *SRY* in therian mammals and the male determining gene in medaka fish [40,41,42], on Y3 chromosome (chromosome 7 in *G. rugosa*) [38], and an unidentified sex-determining gene on Y2 (chromosome 3) (Figure 2d). Thus, we have for the first time found a vertebrate species, which has the sex-chromosomes that include orthologs of the sex-determining genes in mammals, birds and fishes together.

Here, an important question arises—why the three potential sex-chromosomes were involved in the translocations? Was the choice just done at random, or was there some inevitability? The translocations are not reciprocal but triangular, indicating that the three translocations must have occurred simultaneously and been completed at the moment. Therefore, we can speculate that the three chromosome pairs are usually located close to each other in the nucleus. To be so, the three may share a common DNA sequence on each of them, which makes them closely localized to each other, and this makes it possible to join the simultaneously occurring breakages and translocations. Thus, we hypothesize that the breakages on the three chromosomes themselves occurred incidentally, but the three members of chromosomes were chosen nonrandomly; that is, they were inevitably chosen. Nonrandom translocations are still a theory, but a few reports have suggested its real occurrence. In the European brown frog *Rana temporaria*, a translocation occurred between chromosomes 1 and 2 or 1 and 7 in distinct geographic populations, where chromosome 1 was the original sex chromosome, and they are all members of potential sex-chromosomes [43]. A surprisingly similar case is found in a bird, Raso lark *Alauda razae* [44]. The Z chromosome is extended probably by fusions with autosomes 4, 3, and 5: avian Z chromosome is partially homologous to chromosome 1 in ranid frogs (and chromosome 1 in *Xenopus tropicalis*), including Dmrt1, and avian chromosomes 4, 3, and 5 include *AR*, *ME-A1*, and *TYRO3*, respectively, which are located on chromosome 8 of *X. tropicalis* (chromosome 7 of ranid frogs), chromosome 5 of *X. tropicalis* (chromosome 3 of ranid frogs), and chromosome 8 of *X tropicalis* (chromosome 7 of ranid frogs). Thus, chromosomes 3, 4, and 5 fused with the Z chromosome in the bird may be partially homologous to chromosomes 7, 3, and 1, respectively, in ranid frogs, which taxonomically includes the Taiwanese frog.

On the other hand, in some species of vertebrates, repeated sequences comprising a single repeat are often amplified and accumulated on the sex chromosomes bearing different origins. For example, the Bkm sequence, which is abundant in the W chromosomes of snakes, is also found on the Y chromosome of mice [45]. The GATA single repeats are localized on both large Y chromosomes (No. 4) and micro Y chromosomes in phylogenetically different lineages of turtles [46]. The sharing of the same repeated sequences can be explained by translocation of a part of the autosome to the sex chromosomes or vice versa. Another possibility is that they had originally shared the source of the same repeated sequence on their sex chromosomes. If our hypothesis is correct, all six potential sex-determining chromosomes in ranid frogs can form a ring-shaped multivalent if breakage and translocations occur. It was really discovered in the Amazonian frog *Leptodactylus pentadactylus* (2n = 22), which has a dodecavalent at first meiosis comprising six X and six Y chromosomes [19]. It is necessary to compare the sex chromosomes of the Amazonian frog with those of six potential sex-determining chromosomes in ranid frogs by fluorescence in situ hybridization (FISH) painting or gene mapping because the karyotypes are differentiated and visibly different from the common karyotype in ranid frogs.

## 5. Conclusions

The translocations among the three potential sex-determining chromosomes, two of which include orthologs of the sex-determining genes in mammals, birds and fishes, found in this study suggest the nonrandom evolution of multiple sex chromosomes and the existence of a common genomic sequence shared by the potential sex-chromosomes. Identification of the common sequence would provide a hint for understanding the mechanisms of sex-chromosome evolution and turnover among the regular members of homomorphic sex chromosomes.

## Figures and Tables

**Figure 1 cells-10-00661-f001:**
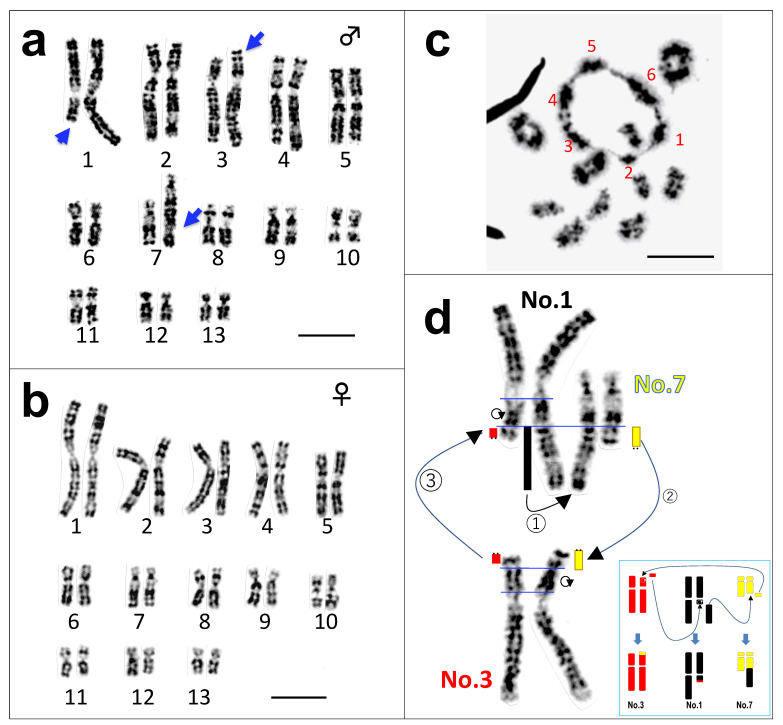
Somatic and meiotic chromosomes of *Odorrana swinhoana*. Late replication banded karyotypes in male (**a**) and female (**b**). Chromosomes 1, 3, and 7 are heteromorphic in males (indicated by arrows), whereas they are homomorphic in females. The metaphase at first meiotic division comprises one hexavalent and ten bivalent rings (**c**). Each of the chromosome members comprising the hexavalent is numbered in red. The presumed pathway of triangular translocations through the chromosomes 1, 3 and 7 in male (**d**). The translocated chromosomal parts are diagrammatically indicated in black (chromosome 1), red (chromosome 3) and yellow (chromosome 7). The three translocations are diagrammatically represented in a box at the right bottom. Telomeres are indicated by two dots. The ring with an arrowhead indicates an inversion. Bar, 10 μm.

**Figure 2 cells-10-00661-f002:**
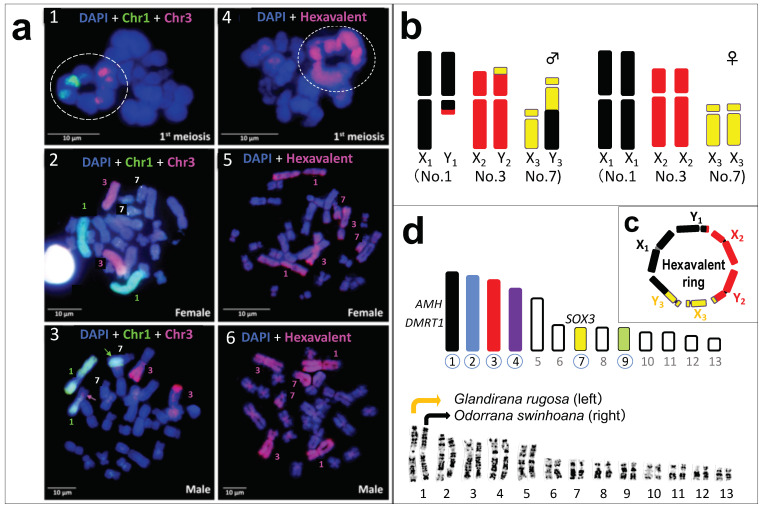
Chromosome painting and diagrams showing the multiple sex-chromosomes and potential sex-chromosomes. Hybridization with probes of microdissected chromosomes 1 (green) and 3 (red) onto the 1st meiotic chromosomes and somatic chromosomes of male and female (**a1**–**a3**). Chromosome 1 probe DNA is shown in green, while chromosome 3, in red. The hybridization with hexavalent probe (red) onto the 1st meiotic chromosomes and somatic chromosomes of male and female (**a4**–**a6**). Hexavalents are boxed by the white dotted line (**a1**,**a4**). Diagrams of six sex-chromosomes in males (left) and females (right) (**b**): chromosomal materials of No. 1, 3 and 7 are shown in black, red and yellow, respectively. The hexavalent is diagrammatically shown in (**c**). Thirteen haploid complements of ranid frogs, of which six potential sex-chromosomes are colored, and their chromosome numbers are circled (upper) (**d**). Orthologs of three sex-determining genes that are mapped in the Japanese frog *G. rugosa* are shown on the left or top of chromosomes 1 and 7, respectively. The late replication banding patterns of 13 chromosome complements are perfectly conserved between *Glandirana rugosa* and *Odorrana swinhoana*, shown at the bottom.

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
