# Peer review of "Evolution of a Multiple Sex-Chromosome System by Three-Sequential Translocations among Potential Sex-Chromosomes in the Taiwanese Frog Odorrana swinhoana"

_cells, 2021, doi:10.3390/cells10030661_

Round 1

Reviewer 1 Report

Translocations between sex-chromosomes and autosomes, generating systems with multiple sex-chromosomes are interesting, as it is not understood whether the sex-chromosomes are involved in fusions more often than expected by chance, or, if so, whether this is caused by selection for linkage between different genes affecting sex determination or with opposing effects on fitness in the two sexes (sexually antagonistic mutations, where polymorphism might become established and select for linkage to the sex-determining locus). The manuscript reports such a case, and is an addition to the many cases already known. The data might become valuable in a future comparative study to test whether sex-chromosomes are involved in fusions unexpectedly often, although a problem is that there is a reporting bias, because researchers report cases involving sex chromosomes, but similar rearrangements involving autosomes are probably reported less often. Intriguingly, the ms states that all ten cases of multiple sex chromosome systems so far reported in amphibians involve fusions of a (homomorphic) sex chromosome with an autosome, and sometimes both the single and multiple-sex chromosome systems coexist within a species or even within the same population, for example, in South and Central 51 American frog species. This suggests that amphibians could be a good system for comparative tests.  

Below, I list several other potential points of interest that emerge from the results, after outlining what I think the observations are. The English in the manuscript is rather hard to understand, but I think that this study describes a system involving 2 Y-A reciprocal translocations, which have generated 2 neo-sex chromosomes in addition to the ancestral XY pair, and I think (based on Figure 2d) that the ancestral karyotype probably had 12 autosome pairs, so the events involved 2 of them (3 and 7 — line 79 mentions chromosome 9, but I think that this is now considered to be incorrect, in light of the new chromosome painting results reported in the ms) and I think that the ancestral XY pair was chromosome 1.

The text should make these basic conclusions clear. At present, these things are not clear, and I am not certain that I understood the observations correctly. Terminology is used that I believe is non-standard (and these terms are not clearly defined), including “three-sided translocations” (in the title), “triangular translocation”. The need for these terms is not clear to me, as the translocations appear fairly standard, other than perhaps the fact that they involve both long and short arms, which might make correct segregation difficult in heterozygotes.  It would be better to use standard terms unless these is something unique about this case.

A valuable aspect of the study is that it identified the telomeres of the chromosomes involved in the trans-locations, and this allowed the study to infer the most likely order of the translocations. It seems that a large part of the long arm of the ancestral Y (chromosome 1), including the terminus was involved in a first exchange with the long arm of chromosome 7, of which two long arm bands near the terminus were transferred to chromosome 1; this was followed by an exchange with the short arm of chromosome 3, which lost one terminal one band, which became moved to the long arm of chromosome 1 (Fig. 1D). In addition, the chromosome 1 long arm and the chromosome 3 short arm each underwent a paracentric inversion.

Other valuable aspects of the data are as follows. The manuscript doesn’t make any of them clear.

  • Multiple sex-chromosome systems are rare in amphibians, so the observations are worth reporting, though other cases have previously been described (including a less detailed study of this population in 1980). Again, the new observations may become valuable for a future comparative study, for example to test whether heteromorphic sex chromosomes might be found in some taxa, while others remain homomorphic, versus heteromorphism being widespread, albeit rare. It should be emphasized that, in the species studied, the heteromorphism is simply due to the translocations, and not to the evolution of size differences between the ancestral chromosome arm and the homologous arm that has become a neo-Y. The text should avoid writing “easy way to quickly achieve heteromorphy of 53 sex chromosomes”, because heteromorphism is not something that is a goal that species achieve, but a change that sometimes happens, and we hope to understand its causes. As the authors mention, foresight is not a property of evolution, so this sentence should be deleted. In the case of translocations, their spread is generally impeded by meiotic problems in heterozygotes, and it seems strange that this is not mentioned, and no observations of crossovers are described (see also below).

  • This example is seen in a population whose most closely related populations do not have the rearrangements, implying that 2 rearrangements involving the sex chromosomes have become fixed in a short evolutionary time. A caveat that should be mentioned is that the results appear to be based on a single sibship (4 males and 3 females), and so they represent only a single sire sampled from the population. The rearrangements might therefore not, in fact, be fixed in the population; however, as they were observed in the 1980 study, and in this set of progeny (which were not selected for having them), it is probably at a high frequency in males.

  • The neo-Y chromosome arms do not appear to be genetically degenerated (based on C-banded karyotypes, which did not detect heterochromatin on any of the three Y chromosomes, and CGH did not detect male or female specific signals). These results suggest that the Y chromosomes have not yet accumulated male-specific repetitive DNA or heterochromatin to any great extent. This again suggests recent origins, or continued crossing over, as repeat accumulation occurs very fast once crossing over stops (though sequencing studies might reveal some accumulation). Importantly, it is also unclear whether the neo-Y chromosome arms undergo crossovers, which would prevent degeneration. This seems a major gap in the study, as there is no expectation that these arms would not continue to recombine unless crossovers in males are strongly restricted to the terminal regions. Can observations of MLH1 foci, or at least of chiasmata be added?

  • If sound evidence were available to show that the rearrangements really did become fixed in a short evolutionary time, this would be potentially interesting, because it would suggesting either that they were favoured by natural selection, or that the population in which they have become established was small, and they fixed by genetic drift. Here, it will be important to discuss whether these translocations might be disadvantageous in heterozygous males, as is generally the case. If so, drift would be less likely. However, the large number of rearrangements, including 2 inversions that are perhaps specific to this population, does suggest the possibility that drift could be responsible, and not selection.

I do not think that it is particularly significant that two out of the three neo-sex chromosomes in the species studied include orthologues of sex determining genes of mammals, birds or fish, and I would question the claim of non-random translocations unless an explicit test has been done.

The claim is repeatedly made in papers that “Sex chromosomes generally evolve from an ordinary autosomal pair“ but this is very vague. It means that Sex chromosomes have repeatedly evolved from ancestral chromosomes without sex-determining genes, either by movement of such genes onto an autosome, or de novo evolution of new genes that have taken over control of sex-determination). These events create X- and Y-linked regions (NOTE: they do NOT create differentiated sex chromosomes) or ZZ-ZW-linked systems, and fusions with autosomes may subsequently occur. If one member of a sex-chromosome pair fuses with an autosome, the number of sex chromosomes is increased, while if both homologues become fused, the size of the sex-chromosome pair is increased. [Here it is unclear why the text discusses fusions, given that the study is about reciprocal translocations]. The latter is the case in placental mammals, including humans, in which an autosome corresponding to kangaroo (marsupial) chromosome 5 was fused with both the X and Y chromosomes (1, 2). To date, the evolution of the translocation sex chromosomes and autosome fusions has been discussed in relation to speciation (3, 4), sexual benefit (5), and Y chromosome degeneration (6, 7). However, in such cases, any possible evolutionary advantages of multiple sex-chromosomes [NOTE that reproductive isolation and speciation  are not advantages that will allow the rearrangements to fix within one species] were acquired after the translocations became fixed and that evolution has no foresight.

Author Response

Dear reviewer 1,

Thank you for the comments and suggestions to our manuscript. We responded all the comments point to point. We put an asterisk on the head of each of our responses and written in blue.

Reviewer 1

Comments and Suggestions for Authors

Translocations between sex-chromosomes and autosomes, generating systems with multiple sex-chromosomes are interesting, as it is not understood whether the sex-chromosomes are involved in fusions more often than expected by chance, or, if so, whether this is caused by selection for linkage between different genes affecting sex determination or with opposing effects on fitness in the two sexes (sexually antagonistic mutations, where polymorphism might become established and select for linkage to the sex-determining locus). The manuscript reports such a case, and is an addition to the many cases already known. The data might become valuable in a future comparative study to test whether sex-chromosomes are involved in fusions unexpectedly often, although a problem is that there is a reporting bias, because researchers report cases involving sex chromosomes, but similar rearrangements involving autosomes are probably reported less often. Intriguingly, the ms states that all ten cases of multiple sex chromosome systems so far reported in amphibians involve fusions of a (homomorphic) sex chromosome with an autosome, and sometimes both the single and multiple-sex chromosome systems coexist within a species or even within the same population, for example, in South and Central 51 American frog species. This suggests that amphibians could be a good system for comparative tests.  

Below, I list several other potential points of interest that emerge from the results, after outlining what I think the observations are. The English in the manuscript is rather hard to understand, but I think that this study describes a system involving 2 Y-A reciprocal translocations, which have generated 2 neo-sex chromosomes in addition to the ancestral XY pair, and I think (based on Figure 2d) that the ancestral karyotype probably had 12 autosome pairs, so the events involved 2 of them (3 and 7 — line 79 mentions chromosome 9, but I think that this is now considered to be incorrect, in light of the new chromosome painting results reported in the ms) and I think that the ancestral XY pair was chromosome 1.

*At present, we do not know which is the ancestral (original) sex chromosome among the three sex chromosomes. Therefore, we could not define the original sex-chromosome in our present study. I understand that the reviewer speculates that chromosome 1 is the original sex chromosome, because this chromosome is often used as a sex chromosome in frog species (please see Miura 2017). However, it is still just a speculation. Our next analysis on SNPs using the frogs from central population bearing no translocations, which is now going on, will identify the original homomorphic sex chromosome in this species.

*We stayed “chromosome 9 “ in L59 because this description is about the paper of Kuramato and he described “chromosome 9”.

The text should make these basic conclusions clear. At present, these things are not clear, and I am not certain that I understood the observations correctly. Terminology is used that I believe is non-standard (and these terms are not clearly defined), including “three-sided translocations” (in the title), “triangular translocation”. The need for these terms is not clear to me, as the translocations appear fairly standard, other than perhaps the fact that they involve both long and short arms, which might make correct segregation difficult in heterozygotes.  It would be better to use standard terms unless these is something unique about this case.

*Ever-known typical translocation is “a reciprocal translocation” between two chromosome pairs.  However, in the case of this frog, the translocations are not reciprocal, but those among three pairs. “Reciprocal” means an exchange of chromosomal parts between two pairs with each other.  On the other hand, in this species, one part of chromosome 1 moved to chromosome 7, but the part of chromosome 7 did not move back to chromosome 1 but to another chromosome 3.  Likewise, from chromosome 3 to chromosome 1.  That’s why we used “three-sided” or “triangular translocations”. Please see our next response below for this argument.

A valuable aspect of the study is that it identified the telomeres of the chromosomes involved in the trans-locations, and this allowed the study to infer the most likely order of the translocations. It seems that a large part of the long arm of the ancestral Y (chromosome 1), including the terminus was involved in a first exchange with the long arm of chromosome 7, of which two long arm bands near the terminus were transferred to chromosome 1; this was followed by an exchange with the short arm of chromosome 3, which lost one terminal one band, which became moved to the long arm of chromosome 1 (Fig. 1D). In addition, the chromosome 1 long arm and the chromosome 3 short arm each underwent a paracentric inversion.

*Thank you for another idea to explain the translocations in this species, which are two reciprocal translocations, 1st between chromosomes 1 and 7, and 2nd between already translocated chromosome 1 and 3. Our idea and reviewer’s idea both give the same figure to form a hexavalent ring at meiosis. However, the latter needs one more chromosome break at the same region on chromosome 1. It is rather harder to be accepted than ours because it requires one more same “by chance”.  In addition, reviewer’s idea requires a prerequisite that the chromosome 1 is the original sex chromosome. If not, the first translocation between 1 and 7 is just the one between two autosome pairs and is not specific to male, giving just polymorphic karyotype and the translocated chromosomes are inherited into both of males and females in the population. This does not promise the translocated karyotypes shown in this study because if second translocation between translocated chromosome 1 and 3 happens, females too should have the translocated 1 or 7. As mentioned above, we do not have any evidence about which is the original sex chromosome in this species. Thus, we are sorry not for accepting the reviewer’s idea of two reciprocal translocations at present.

Other valuable aspects of the data are as follows. The manuscript doesn’t make any of them clear.

  • Multiple sex-chromosome systems are rare in amphibians, so the observations are worth reporting, though other cases have previously been described (including a less detailed study of this population in 1980). Again, the new observations may become valuable for a future comparative study, for example to test whether heteromorphic sex chromosomes might be found in some taxa, while others remain homomorphic, versus heteromorphism being widespread, albeit rare. It should be emphasized that, in the species studied, the heteromorphism is simply due to the translocations, and not to the evolution of size differences between the ancestral chromosome arm and the homologous arm that has become a neo-Y. The text should avoid writing “easy way to quickly achieve heteromorphy of 53 sex chromosomes”, because heteromorphism is not something that is a goal that species achieve, but a change that sometimes happens, and we hope to understand its causes. As the authors mention, foresight is not a property of evolution, so this sentence should be deleted. In the case of translocations, their spread is generally impeded by meiotic problems in heterozygotes, and it seems strange that this is not mentioned, and no observations of crossovers are described (see also below).

 *We do not mean a goal of sex chromosome to be heteromorphic. Sex chromosome tends to be heteromorphic from homomorphic in many species. If translocation happens, it brings about heteromorphic chromosomes very quickly and easily.  That’s why we mentioned “might be an easy way to achieve heteromorphy of sex chromosomes”. This does not mean the goal, but one direction.

  • This example is seen in a population whose most closely related populations do not have the rearrangements, implying that 2 rearrangements involving the sex chromosomes have become fixed in a short evolutionary time. A caveat that should be mentioned is that the results appear to be based on a single sibship (4 males and 3 females), and so they represent only a single sire sampled from the population. The rearrangements might therefore not, in fact, be fixed in the population; however, as they were observed in the 1980 study, and in this set of progeny (which were not selected for having them), it is probably at a high frequency in males.

* I think this is misunderstanding of reviewer for our result. The 4 males and 3 females were collected from Northern population 1 (New Taipei) at different dates, and they are all wild frogs.  On the other hand, 5 males and 5 females (F1) are sib-ship from the same parents, who are each one from the 4 males and 3 females. The results on F1 males and females clearly show that the translocations are inherited to next generation with no troubles in segregation of 3 X chromosomes and 3 Y chromosomes at the meiosis. And, we actually asked Dr. Kuramoto who discovered this multiple sex chromosomes in this species.  He investigated the frogs from some locations including northern, central and eastern regions, and the males had all the translocation, which is also mentioned in his paper. Thus, we are thinking that unknown drastic change in nature occurred during the past 40 yeas to reduce the translocated chromosomes from the populations. We would like to investigate the evolutionary reasons if any in near future.

  • The neo-Y chromosome arms do not appear to be genetically degenerated (based on C-banded karyotypes, which did not detect heterochromatin on any of the three Y chromosomes, and CGH did not detect male or female specific signals). These results suggest that the Y chromosomes have not yet accumulated male-specific repetitive DNA or heterochromatin to any great extent. This again suggests recent origins, or continued crossing over, as repeat accumulation occurs very fast once crossing over stops (though sequencing studies might reveal some accumulation). Importantly, it is also unclear whether the neo-Y chromosome arms undergo crossovers, which would prevent degeneration. This seems a major gap in the study, as there is no expectation that these arms would not continue to recombine unless crossovers in males are strongly restricted to the terminal regions. Can observations of MLH1 foci, or at least of chiasmata be added?

* This reviewer’s comment is based on other vertebrates such as mammals, birds, reptiles and fishes. In anuran amphibians such as evolved frogs including Rana, Hyla, Bufo and so on, the bivalent always forms a ring with chiasma or fusion restricted to both terminal tips at 1st meiosis. Please see the classic review of Molescalchi (1973). In these frog groups, 94% species have homomorphic sex chromosomes, but they do not accumulate any Y specific DNA. No recombination almost along the chromosomes at male meiosis gives no accumulation of male specific DNA.  Likewise, the hexavalent observed in this species does not drive any DNA specific accumulation caused by the paring figure (end to end fusion with no chiasmata) or even by heteromorphic chromosomes. Thus, we concluded its very recent origin.

  • If sound evidence were available to show that the rearrangements really did become fixed in a short evolutionary time, this would be potentially interesting, because it would suggesting either that they were favoured by natural selection, or that the population in which they have become established was small, and they fixed by genetic drift. Here, it will be important to discuss whether these translocations might be disadvantageous in heterozygous males, as is generally the case. If so, drift would be less likely. However, the large number of rearrangements, including 2 inversions that are perhaps specific to this population, does suggest the possibility that drift could be responsible, and not selection.

*Thank you for this comment. We would like to challenge this issue at our next step. For this, we need to investigate more populations across the island. As mentioned above, 40 years ago, the all male frogs investigated seem to have had the translocations if Kuramoto said the truth. Which happened, selection or genetic drift, it is necessary to be solved by carefully analyzing many populations.

I do not think that it is particularly significant that two out of the three neo-sex chromosomes in the species studied include orthologues of sex determining genes of mammals, birds or fish, and I would question the claim of non-random translocations unless an explicit test has been done.

*Before this debate, I need to talk about nonrandom turnover of sex chromosomes in frogs.  In the reviews of Miura (2007 and 2017), nonrandom turnover was just a theory but Jefferies et al. investigated 28 frog species and identified their sex chromosomes and proved the theory is the case. They found that the frogs repeated turnover of sex chromosomes among five members of the chromosomes.  Thus, non random turnover is the case. And which supports our theory in this paper that the triangular translocations in this species occurred at non-random because the three chromosomes involved in the translocations all belong to the members of the potential sex-chromosomes. In 1999, it was proved that mammalian sex chromosomes and avian sex chromosomes do not share their origins but they are completely different chromosomes and evolved independently. The sex chromosomes identified in other vertebrates often include synteny of mammalian sex chromosomes or that of avian sex chromosomes, but not both together.  Our investigation for the first time found a species who has Y chromosomes that are homologous or partially to both of mammalian and avian sex chromosomes.  We will investigate which gene works as a male determining gene, Dmrt1 on Y1 or Sox3 on Y3, from now on.  We are sure this finding is unique and very exciting.

The claim is repeatedly made in papers that “Sex chromosomes generally evolve from an ordinary autosomal pair“ but this is very vague. It means that Sex chromosomes have repeatedly evolved from ancestral chromosomes without sex-determining genes, either by movement of such genes onto an autosome, or de novo evolution of new genes that have taken over control of sex-determination). These events create X- and Y-linked regions (NOTE: they do NOT create differentiated sex chromosomes) or ZZ-ZW-linked systems, and fusions with autosomes may subsequently occur. If one member of a sex-chromosome pair fuses with an autosome, the number of sex chromosomes is increased, while if both homologues become fused, the size of the sex-chromosome pair is increased. [Here it is unclear why the text discusses fusions, given that the study is about reciprocal translocations]. The latter is the case in placental mammals, including humans, in which an autosome corresponding to kangaroo (marsupial) chromosome 5 was fused with both the X and Y chromosomes (1, 2). To date, the evolution of the translocation sex chromosomes and autosome fusions has been discussed in relation to speciation (3, 4), sexual benefit (5), and Y chromosome degeneration (6, 7). However, in such cases, any possible evolutionary advantages of multiple sex-chromosomes [NOTE that reproductive isolation and speciation  are not advantages that will allow the rearrangements to fix within one species] were acquired after the translocations became fixed and that evolution has no foresight.

Comment 1:

“Sex chromosomes generally evolve from an ordinary autosomal pair“ but this is very vague.

*The sentence was not enough to explain a sex chromosome and thus we revised as follows in L33-34 :

“Sex chromosomes generally evolve from an ordinary autosomal pair acquiring a sex-determining gene and thus are composed of X and Y chromosomes in an XX-XY system, or Z and W chromosomes in a ZZ-ZW system.”

Reviewer’s comment 2:

‘Here it is unclear why the text discusses fusions, given that the study is about reciprocal translocations’

*Translocation between autosome and sex chromosome does not always mean a reciprocal translocation. Often, whole autosome is fused with X or Y chromosome, or with both.  Thus, we used “fusion” here. As mentioned above, it is plausible that triangular translocations but not two reciprocal translocations occurred in this species.

Reviewer’s comment 3:

‘reproductive isolation and speciation  are not advantages that will allow the rearrangements to fix within one species’

*This sentence is based on the description of the reference paper cited here and is not my own idea.

Reviewer 2 Report

Dear authors.

It's always nice to see chromosomes and the conjunction of "obsolete" techniques (such as replication bands) with FISH or CGH, at least to me. I've always been fascinated on how diverging apparently well preserved karyotypes (such as frogs) can really be after the correct techniques are applied.

The quality of your plates and figures is very good, the description and overall writing here is very good (although I have some minor comments bellow) and the results are very interesting (to me). I am a bit skeptical on the "inetability" of these events and that these may have occurred simultaneously, though. Nevertheless, as you clearly stated this is specullative and it's written in such a way that it's inside my confort area.

I got some suggestions/aditional points:

1) As I can see from Fig. S2, there is some heterochromatin at least on some of the putative sexual pairs (by the way, there is actually some heterochromatin at the centromeres. I'd suggest you to change the legend to "No visible non-centromeric heterochromatin was observed on the three Y chromosomes"). Could these be somehow related to the translocation events, as suggested in the lasta paragraph of the discussion? Could authors add even one sentence about this fact?

2) Similarly, there are some interstitial telomeric sequences on some of the chromosomes showed at Fig. S2 (outside from the telomeric signals at pericentric positions). I guess this is but signal noise, but just wanted to doublecheck whether these could be real or not.

3) L235-243 Authors state here that this Taiwanese frog holds three sex determination candidate genes, one on each putative Y chromosome, based on reference 21. As it’s written here, though, it looks (to me), like this is merely speculation from the authors (which is not the case, of course). Could authors possibly rephrase L234-235 so that this fact remains clearly and implicitly stated here?

Some other minor typos bellow:

L25        Ortologue genes, I guess.

L49        There is a space missing in between “(13–20).The”

L56         Please, change “between the two chromosomes, No 1 and No 9” to “between chromosomes No 1 and No 9

L87         Please, change “basically were” to “were basically”

L106      “slides were incubated at 65 °C for 2 h and in 70% formamide for 2 min in 70% ethanol” This sentence requires a bit of rephrasing to be easier to understand

L108      “Hybridization solution” without capital letter at the beginning of the word.

L108-109             “2xSSC: distilled water: BSA (bovine serum albumin): 50% Dextran= 1:1:1:2”. Why so many “:” here?

L110      “Slides washed in 50% formamide for 20 min and rinsed 10 times, in 2xSSC they were rinsed 15 times and incubated for 15 min, in 1xSSC they were rinsed 15 times and incubated 15 min, and in 4xSSC they were incubated for 5 min and rinsed 15 times” Could authors possibly rephrase this?

L127-129             “before 10 μL Hybridization solution (2xSSC SSC: DW: BSA: 50% Dextran= 1:1:1:2) was dropped onto slides with 10 μL probes.”             “Hybridization solution” without capital letter at the beginning of the word. Also this sentence needs a bit of rephrasing (not “was” but “were”; It’s a bit complicated to read as it is…)

L137      There is an space missing in between “(23)(Table 1)” 

Thus, I recommend this manuscript to undergo minor revissions.

Best regards

Author Response

Dear reviewer 2,

Thank you for the comments and suggestions to our manuscript. We responded all the comments point to point. We put an asterisk on the head of each of our responses and written in blue.

Reviewer 2

Dear authors.

It's always nice to see chromosomes and the conjunction of "obsolete" techniques (such as replication bands) with FISH or CGH, at least to me. I've always been fascinated on how diverging apparently well preserved karyotypes (such as frogs) can really be after the correct techniques are applied.

The quality of your plates and figures is very good, the description and overall writing here is very good (although I have some minor comments bellow) and the results are very interesting (to me). I am a bit skeptical on the "inetability" of these events and that these may have occurred simultaneously, though. Nevertheless, as you clearly stated this is specullative and it's written in such a way that it's inside my confort area.

I got some suggestions/aditional points:

1) As I can see from Fig. S2, there is some heterochromatin at least on some of the putative sexual pairs (by the way, there is actually some heterochromatin at the centromeres. I'd suggest you to change the legend to "No visible non-centromeric heterochromatin was observed on the three Y chromosomes"). Could these be somehow related to the translocation events, as suggested in the lasta paragraph of the discussion? Could authors add even one sentence about this fact?

*Thanks.  This comment is right. The three Y chromosomes have centromeric or pericentromeric heterochromatin, but do not show any Y-specific heterochromatin.  Therefore, we added “Y-specific” to the legend sentences in Fig. S2 and text (L218) as follows :

“C-banded karyotypes of male (A) and female (B). Arrows indicate three Y-chromosomes. No stained Y-specific heterochromatin was observed on the three Y chromosomes. Bar, 10 μm.”

“No stained Y-specific heterochromatin was detected on the three Y chromosomes (Fig. S2).”

*We think that heterochromatin is not involved in the translocations because no heterochromatin is located on or close to the breakpoints of the translocations.

2) Similarly, there are some interstitial telomeric sequences on some of the chromosomes showed at Fig. S2 (outside from the telomeric signals at pericentric positions). I guess this is but signal noise, but just wanted to doublecheck whether these could be real or not.

*Basically, we observed no visible Y-specific heterochromatin on the three Y chromosomes. If any, they would be detected by CGH. However, CGH did not show any male specific signals.  Thus, we concluded that no male specific DNA specialization occurs yet on the three Y chromosomes, and this indicates that their origin is very young.

*The telomere mapping painted centromeric and telomeric regions. This happens in other species too because the centromeric heterochromatin includes the similar sequence to telomeres.  In our study, this centromeric staining does not matter, because we used this technique to investigate whether the terminal ends or internal region of the translocated chromosomes include telomeres or not and their positions.

3) L235-243 Authors state here that this Taiwanese frog holds three sex determination candidate genes, one on each putative Y chromosome, based on reference 21. As it’s written here, though, it looks (to me), like this is merely speculation from the authors (which is not the case, of course). Could authors possibly rephrase L234-235 so that this fact remains clearly and implicitly stated here?

*As the comment, our explanation was short.  Thus, we added the sentences in L254-257:

The karyotypes of 2n=26 in true frogs are highly conserved with little chromosome rearrangements [10, 21, 25]. In fact, the 13 chromosome complements are almost perfectly conserved between G. rugosa and O. swinhoana based on late replication banding pattern (Fig. 2d).”

*In addition, we added “may” in the sentence in L258, and “in G. rugosa” in L261 and 263

Some other minor typos bellow:

L25        Ortologue genes, I guess.

*“Orthologue” is correct.

L49        There is a space missing in between “(13–20).The”

*I think no space should be here.

L56         Please, change “between the two chromosomes, No 1 and No 9” to “between chromosomes No 1 and No 9

*I would like to emphasize “two chromosomes” here in L59.  Therefore, put the comma after the words and then “No.1 and No.9” were followed.

L87         Please, change “basically were” to “were basically”

*“were basically…” is described there.

L106      “slides were incubated at 65 °C for 2 h and in 70% formamide for 2 min in 70% ethanol” This sentence requires a bit of rephrasing to be easier to understand

*Very sorry for this section.  The descriptions were confusing, and thus we revised the whole parts of the sections of telomere mapping and CGH as follows in L109-139:

Previously described methods [46, 47] were followed to determine the chromosomal locations of telomeric (TTAGGG)n sequences. We used biotin-16-UTP labeled TTAGGG repeats as the prove. For hybridization of biotin-labeled probes to Odorrana swinhoana chromosomes, slides were first incubated at 65 °C for 2 h in 70% formamide and were rinsed in 70% ethanol for 5 min, in 100% ethanol for 5 min and dried. Then, 10 μl hybridization solution containing 50% formamide, 10% dextran sulfate, 2xSSC, 40mM sodium phosphate and 1x Denhardt’s solution were dropped onto the slides with 10 μl probes under a coverslip. After the slides were incubated at 37 °C overnight, they were washed in 50% formamide for 20 min and rinsed in 2xSSC for 15 min, in 1xSSC for 15 min, and in 4xSSC for 5 min. After completed the wash and rinse, the slides were stained with avidin labeled with fluorescein isothiocyanate (Avidin-FITC; Invitrogen, CA, USA). After incubation at 37 °C for 1 h, slides were rinsed in 4xSSC and washed on a shaker for 10 min, then in 0.1% Nomident/4xSSC (NP-40) for 10 min, in 4xSSC for 5 min and finally in 2xSSC for 5 min. Slides were subsequently stained with 1 μg/ml 4,6-diamidino-2phenylindole (DAPI). Fluorescence hybridization signals were captured using a cooled CCD camera mounted on a ZEISS microscope (Zeiss, Germany) and processed using MetaSystems ISIS v.5.0. software (MetaSystems, Alltlussheim, Germany).

 CGH was performed to detect genomic sequence differences between males and females. An amount of 500 ng of male and 500 ng of female genomic DNA were labeled with biotin-dUTP and Cy3 dUTP, respectively, using a Nick translation Kit (Sigma Aldrich) by incubating at 15 ℃ for two hours and heating at 65 ℃ for 10 min. After ethanol precipitation, the slides were denatured by heating at 75 °C for 10 min before 10 μl hybridization solution containing 50% formamide, 10% Dextran sulfate, 2xSSC, 40mM sodium phosphate and 1x Denhardt’s solution was dropped onto slides with 10 μl probes. Slides were incubated at 37 °C for 3–4 days in a humid box. Hybridized slides were washed in 4xSSC for 7 min, in NP-40/4xSSC for 7 min, and in 4xSSC for 7 min, and by shaking in 2xSSC for 5 min. Slides were subsequently stained with 1 μg/ml 4,6-diamidino-2phenylindole (DAPI).

***The comments below are also about these two sections. So, please see the above revised sentences.

L108      “Hybridization solution” without capital letter at the beginning of the word.

*Revised as above.

L108-109             “2xSSC: distilled water: BSA (bovine serum albumin): 50% Dextran= 1:1:1:2”. Why so many “:” here?

*Revised as above.

L110      “Slides washed in 50% formamide for 20 min and rinsed 10 times, in 2xSSC they were rinsed 15 times and incubated for 15 min, in 1xSSC they were rinsed 15 times and incubated 15 min, and in 4xSSC they were incubated for 5 min and rinsed 15 times” Could authors possibly rephrase this?

*Revised as above.

L127-129             “before 10 μL Hybridization solution (2xSSC SSC: DW: BSA: 50% Dextran= 1:1:1:2) was dropped onto slides with 10 μL probes.”             “Hybridization solution” without capital letter at the beginning of the word. Also this sentence needs a bit of rephrasing (not “was” but “were”; It’s a bit complicated to read as it is…)

L137      There is an space missing in between “(23)(Table 1)” 

*We have inserted a space there.

Round 2

Reviewer 1 Report

The English remains too poor to understand properly. The term “triangular translocation” is still used, but it is non-standard and does not convey a clear meaning. I was forced to edit the text myself in order to understand it, and I provide it below. It is much shorter than the original, which is appropriate, as the study is purely descriptive, and describes a new instance of a well-known type of chromosome rearrangement. It is interesting that 3 reciprocal translocations occurred in a short evolutionary time. It is less interesting (but worth reporting) that the regions that became attached to sex-linked regions have not degenerated, as this is what one expects, given that they presumably continued to recombine. Unfortunately, the study appears not to have examined MLH1 foci, so crossovers cannot be assessed. Other than this weakness, it seems technically sound, and the observations should be recorded.

It seems far-fetched to suggest significance of the fact that the chromosomes involved carry orthologues of the sex-determining genes in mammals, birds or fish. With 13 chromosomes, it is highly likely that this will be the case. The authors have not shown that there is an unexpected representation of such genes on this chromosome, with an adequate statistical test, taking proper account of the numbers of such genes and the chance that some of them will be carries on one of these chromosomes. Unless chance has been excluded, it is not “an important question …why the three potential sex-chromosomes were involved in the triangular translocations”. I therefore recommend deleting this from the ms, including the Discussion, which contains little else.

EDITED TEXT

Sex chromosomes may evolve after an autosomal pair acquire a sex-determining gene. In some cases, recombination has then become suppressed, leading to evolution of a differentiated X Y or ZW chromosome pair.  Sometimes one member of this pair then becomes fused with an autosome, generating a multiple sex chromosome system with a so-called “neo- sex-chromosome”. If either member of a sex-chromosome pair is fused with an autosome, the number of sex chromosomes is increased, while if both homologues become fused, the number of sex chromosomes pairs remains the same but the sex chromosome sizes are larger than in the ancestor.

It is unclear what the authors mean by “the evolutionary meaning of the translocation between sex chromosomes and autosomes has been documented in …”. I think their meaning is something like the following “translocations between sex chromosomes and autosomes have been studied in relation to speciation [3, 4], sexually antagonistic selection [5], and the degeneration of Y chromosomes [6, 7]. However, these changes occur after translocations happen, and do not help understand why they happen in the first place.

In amphibians, multiple sex-chromosome situations are very rare, with approximately ten cases reported to date, all of which involve initially homomorphic sex chromosomes, with fusions creating heteromorphic pairs [13–20]. In a few species, both the single and multiple-sex chromosome systems coexist within a species or even within the same population, for example, in South and Central 53 American frog species [15–17].

Here, we describe a fusion system in the Taiwanese brown frog, Rana narina (a synonym for Odorrana swinhoana). The first report of multiple sex chromosomes in amphibians was a male-specific translocation between chromosomes 1 and 9 of the species in 1980 by Kuramaoto [13]. The male-specificity suggests that the translocation occurred between the two members of the sex chromosomes, forming a ♂X1Y1X2Y2- ♀X1X1X2X2 system. The sex chromosomes in amphibians undergo turnovers during speciation or the differentiation of geographic populations within species. Turnovers involving six chromosomes are over-represented (1–4, 7, and 9 of 13 the haploid complement) [21, 22]. It could therefore be significant that chromosomes 1 and 9 are involved in the translocation in the Taiwanese frog. The identification of the chromosomes was, however, uncertain because the chromosomes were stained with only a single Giemsa 70 solution, and the identification was based on their size and shape. 71

In this study, to reliably identify the chromosomes involved in the male-specific translocation, we re-investigated the somatic chromosomes as well as meiotic chromosomes of this frog species using chromosome banding and molecular mapping techniques. Unexpectedly, the translocation was found not to be a single but a triple one.

  1. Materials and Methods \

I DID NOT EDIT THIS SECTION

Results and Discussion

3.1. Three heteromorphic sex chromosomes in males 143 To re-examine the conclusions of the previous studies [13, 14], we investigated karyotypes of 144 4 males and 3 females collected from the northern population (New Taipei), Taiwan, using 145 late replication banding [23] (Table S1). All the females showed homomorphic pairs of 13 146 haploid complements (2n = 26), while all the males had three pairs of heteromorphic chromosomes, which were Nos. 1, 3, and 7 (Fig. 1a and b). In the male karyotype, large part of 148

4 of 10

the long arm of chromosome 1 was missing, whereas the long arm of chromosome 7 was 149 much longer than the homologue. In addition, the short arm of chromosome 3 was slightly 150 longer than the homologue (Fig.1 and Fig.S1a). We confirmed that the three heteromorphic chromosomes were transmitted to the male offspring in one brood of their F1 generation (Table S1). These heteromorphic chromosomes suggest that more than one translocation occurred, not conforming to the results of Kuramoto [13, 14].

COMMENT: Here, it would be helpful to provide a diagram to make clear the events that are inferred to have resulted in the situation observed. Figure 2c can be explicitly mentioned, so that readers know that this gives an explanation. The term “triangular translocation” is still used, but it is non-standard and does not convey a clear meaning. The production of a ring hexavalent is clear, and it is unhelpful to use words that make the meaning less clear.

3.2. A hexavalent ring at male meiosis 165 To investigate the number of translocations, we observed paring figures of the first 166 meiotic chromosomes in male testes. It was found that the meiotic karyotype comprised 167 one large ring-shaped hexavalent, together with ten ring bivalents at the first meiotic metaphase (Fig. 1c and Fig.S1b, c). Three chromosome pairs must therefore be involved in the translocations. In addition, we detected telomeres in all the chromosomes involved in the 171 translocations (Fig.S1c, d). Therefore, the most likely, expected order of translocations to 172 form the hexavalent ring is as follows: the large part of the long arm of chromosome 1, 173 including the terminal tip, was translocated to the long arm of chromosome 7, in which 174

5 of 10

the terminal tip (including two bands) of long arm was broken before the fusion and 175 moved to the short arm of chromosome 3, one band of whose short arm terminus was broken before the fusion and moved to the long arm of chromosome 1, which lost the large part of the long arm (Fig. 1d). In the remaining parts of the chromo- 178 some 1 long arm and of the chromosome 3 short arm, one paracentric inversion also occurred 179 (Figs. 1d and S1a).

181 3.3. Direct proof of the chromosome members involved in triangular translocations 182 To directly identify the chromosome members involved in the triangular transloca- 183 tions, we performed hybridization painting using the microdissected chromosomal 184 DNA probes. Both of the chromosome 1 and 3 probes (green and red, respectively) 185 painted parts of the hexavalent (Fig. 2a-1). COMMENT: Here, the next sentences appear to be garbled. I attempted to guess the meaning, but the authors need to check that it is correct. The chromosome 1 probe painted chromosome pair 1 in the female (Fig.2a-2), and also the heteromorphic chromosome pair 1 and the long arm of the longer chromosome 7 in male (green arrow in Fig.2a-3). The chromosome 3 probe painted chromosome pair 3 in females , as expected(Fig.2a-2), and also the heteromorphic chromosome pair 3 and the short arm of the shorter chromosome 1 in males (pink arrow in Fig. 2a-3). These results directly prove that chromosomes 1 and 3 were involved in the male specific translocations from chromosome 1 to 7 and from chromosome 3 to 1. Next, hybridization using the hexavalent DNA probe painted chromosome pairs 1, 3 193 and 7 in female, while heteromorphic chromosome pairs 1, 3 and 7 in male (Fig.2a-4, 5, 194 6). Thus, it is concluded that the male-specific triangular translocations among chromo- 195 somes 1, 3 and 7 created a system of six sex-chromosomes, ♂X1Y1X2Y2X3Y3 - 196 ♀X1X1X2X2X3X3 (Fig.2b and c). 197 198

199 Figure 2c.

3.4. No male DNA specialization on the three Y chromosomes 216 To examine the extent of chromosomal specialization of the three Y chromosomes, 217 we observed C-banded karyotypes. No Y-specific heterochromatin was detected 218 on the three Y chromosomes (Fig. S2). In addition, complementary genomic hybridization (CGH) on the male and female chromosomes detected neither male nor female specific signals (Fig. S3). These results show that the three Y chromosomes have not yet accumulated male-specific DNA or heterochromatin.

3.5. Population variation in the sex-chromosome system 224 To elucidate the sex-chromosomal variations within this species, we observed the karyotypes of the frogs collected from two other populations (Table S1), of which one 226 was located just south of the first population (New Taipei), and the other was around the central region of the island (Nantou Ren Ai). We observed no translocations or heteromorphic sex-chromosomes in the males or females based on the late replication and C-banding patterns (Figs S4–6). This result suggests that the translocations in males are restricted in some geographic populations of the species and originated very recently.

Author Response

Dear reviewer,

Thank you for your comments.

We described our responses to the comments point to point in purple, and revised our manuscript (also in purple in the text) according to the comments in the cases we agreed to.  On the other hand, we also stated the reason why we requested to stay our sentences with no changes.

Comments and Suggestions for Authors

1The English remains too poor to understand properly. The term “triangular translocation” is still used, but it is non-standard and does not convey a clear meaning. I was forced to edit the text myself in order to understand it, and I provide it below. It is much shorter than the original, which is appropriate, as the study is purely descriptive, and describes a new instance of a well-known type of chromosome rearrangement. It is interesting that 23 reciprocal translocations occurred in a short evolutionary time. It is less interesting (but worth reporting) that the regions that became attached to sex-linked regions have not degenerated, as this is what one expects, 3given that they presumably continued to recombine. Unfortunately, the study appears not to have examined MLH1 foci, so crossovers cannot be assessed. Other than this weakness, it seems technically sound, and the observations should be recorded.

1 The manuscript was written through reviewing of all the authors and finalized by English editing by a company.  We have attached the certificate.

2 The translocations in this species are not reciprocal ones.  As we mentioned in our last response, “reciprocal” means exchange of a chromosomal part between two different chromosomes.  In this species, a part of chromosome 1 was translocated to chromosome 7, and from 7 to 3 (not to 1) and from 3 to 1 (not to 7).  That’s why we used triangular translocations. However, according to the comment by reviewer that it is hard to be understood, we changed the word to “three or three sequential” translocations in the title and text.

3 As we mentioned in our last response, recombination in male 1st meiosis in these frog groups usually does not occur except at terminal region and thus the chromosomes basically form a ring. In fact, we showed the meiotic metaphases in Fig.1 and Fig.S1, indicating one hexavalent ring and 10 bivalent ring (all form a ring).  Therefore, if we examined MLH1 foci in the meiosis, signals would be restricted around terminal regions, as shown in Amazonian frog with a multivalent ring (10X and 10Y chromosomes; Noronha RCR 2020 (Sci rep)). At a next step of our study, we would like to challenge MLH1 foci analysis to discriminate an end to end fusion from recombination around the terminal regions of the hexavalent, because end to end fusion could affect degeneration of the terminal regions on the Y chromosomes owning to reduced or no recombination.

It seems far-fetched to suggest significance of the fact that the chromosomes involved carry orthologues of the sex-determining genes in mammals, birds or fish. With 13 chromosomes, it is highly likely that this will be the case. The authors have not shown that there is an unexpected representation of such genes on this chromosome, with an adequate statistical test, taking proper account of the numbers of such genes and the chance that some of them will be carries on one of these chromosomes. Unless chance has been excluded, it is not “an important question …why the three potential sex-chromosomes were involved in the triangular translocations”. I therefore recommend deleting this from the ms, including the Discussion, which contains little else.

Dmrt1 and Amh are proved to be mapped on chromosome 1 and Sox3 be on chromosome 7 in G. rugosa (Uno et al. 2008; Kodama et al. 2015).  In addition, karyotypes are highly conserved in true frogs with little rearrangements (Miura 1995).  Please see Fig. 2d; particularly, the late replication banding patterns are almost perfectly conserved between G. rugosa and this frog species. The conservation of chromosomes 1-4 and 7 in true frogs is also shown in the study of Jefferies et al. (2018).

Based on these studies and our indirect evidence in Fig.2d, we are saying that the three genes are located on two Y chromosomes in this frog species. We showed indirect evidence for the mapping, but they promise the conservation of location of the three genes. We found for the first time a species who has sex chromosomes together including orthologues of mammalian, avian and fish (platypus) sex determiming genes. We would like to emphasise this finding in our manuscript and hope reviewer and editor to consider this point. Therefore, please let us stay the sentences about three genes on the Y chromosomes.

The two paragraphs in L268-308 describe our speculation for the translocations among potential sex chromosomes. These paragraph are necessary for this manuscript because the important finding is translocations among three potential sex chromosomes, and which suggests nonrandom choice of the three chromosomes involved in translocations. We are saying that breakage and translocations may have occurred by chance, but the choice of chromosomes may be at nonrandom. Thus, we presented other two similar examples in frogs, and one similar case in bird. We are now investigating turnover of sex chromosomes in Japanese frog, in which we again found two turnovers among the three potential sex chromosomes 1, 3 and 7 (unpublished), suggesting some genetic relationships such as cross talk of gene expression during sex determination, or genetic affinity for sex chromosome turnover and evolution among the three chromosomes. We therefore request to stay these paragraphs.

EDITED TEXT

Sex chromosomes may evolve after an autosomal pair acquire a sex-determining gene. In some cases, recombination has then become suppressed, leading to evolution of a differentiated X Y or ZW chromosome pair.  Sometimes one member of this pair then becomes fused with an autosome, generating a multiple sex chromosome system with a so-called “neo- sex-chromosome”. If either member of a sex-chromosome pair is fused with an autosome, the number of sex chromosomes is increased, while if both homologues become fused, the number of sex chromosomes pairs remains the same but the sex chromosome sizes are larger than in the ancestor.

Thank you for the “after”. We added it to our sentence and also added “ a pair of” in L33-35.

“either a homologue of ….” was revised to “either homologue of …” in L37.

For the other parts, please stay as we wrote because we followed our English editing by a  company.

It is unclear what the authors mean by “the evolutionary meaning of the translocation between sex chromosomes and autosomes has been documented in …”. I think their meaning is something like the following

“translocations between sex chromosomes and autosomes have been studied in relation to speciation [3, 4], sexually antagonistic selection [5], and the degeneration of Y chromosomes [6, 7]. However, these changes occur after translocations happen, and do not help understand why they happen in the first place.

Not “studied” but “documented or discussed” because this is discussion but not result.

“changes” is not clear for us.  Please let us stay our sentences.

In amphibians, multiple sex-chromosome situations are very rare, with approximately ten cases reported to date, all of which involve initially homomorphic sex chromosomes, with fusions creating heteromorphic pairs [13–20]. In a few species, both the single and multiple-sex chromosome systems coexist within a species or even within the same population, for example, in South and Central 53 American frog species [15–17].

Here, we describe a fusion system in the Taiwanese brown frog, Rana narina (a synonym for Odorrana swinhoana). The first report of multiple sex chromosomes in amphibians was a male-specific translocation between chromosomes 1 and 9 of the species in 1980 by Kuramaoto [13]. The male-specificity suggests that the translocation occurred between the two 1members of the sex chromosomes, forming a ♂X1Y1X2Y2- ♀X1X1X2X2 system. The sex chromosomes in amphibians undergo turnovers during speciation or the differentiation of geographic populations within species. 2Turnovers involving six chromosomes are over-represented (1–4, 7, and 9 of 13 the haploid complement) [21, 22]. It could therefore be significant that chromosomes 1 and 9 are involved in the translocation in the Taiwanese frog. The identification of the chromosomes was, however, uncertain because the chromosomes were stained with only a single Giemsa 70 solution, and the identification was based on their size and shape. 71

In this study, to 3reliably identify the chromosomes involved in the male-specific translocation, we re-investigated the somatic chromosomes as well as meiotic chromosomes of this frog species using chromosome banding and molecular mapping techniques. Unexpectedly, the translocation was found not to be a single but a triple one.

1 Not two members of “sex chromosomes”, but of “potential sex or sex determining chromosomes”.

The meaning should not to be neglected.  “Potential” means that it is just autosome but can be a sex chromosome in case of turnover.

2 “Turnovers involving sex chromosomes” reminds us of six times turnover among six chromosomes.

3 “reliably” is good, but is not so different from “precisely”. 

As mentioned above, please stay the last sentence for sex determining orthologues.

  1. Materials and Methods \

I DID NOT EDIT THIS SECTION

Results and Discussion

3.1. Three heteromorphic sex chromosomes in males 143 To 1re-examine the conclusions of the previous studies [13, 14], we investigated karyotypes of 144 4 males and 3 females collected from the northern population (New Taipei), Taiwan, using 145 late replication banding [23] (Table S1). All the females showed homomorphic pairs of 13 146 haploid complements (2n = 26), while all the males had three pairs of heteromorphic chromosomes, which were Nos. 1, 3, and 7 (Fig. 1a and b). In the male karyotype, large part of 148

4 of 10

the long arm of chromosome 1 was missing, whereas the long arm of chromosome 7 was 149 much longer than the homologue. In addition, the short arm of chromosome 3 was slightly 150 longer than the homologue (Fig.1 and Fig.S1a). We confirmed that the three heteromorphic chromosomes were transmitted to the male offspring in one brood of their F1 generation (Table S1). These heteromorphic chromosomes suggest that more than one translocation occurred, not conforming to the results of Kuramoto [13, 14].

1 We did not “re-examine the conclusion” but confirmed the results of previous studies, because we completely agreed to the conclusion.

COMMENT: Here, it would be helpful to provide a diagram to make clear the events that are inferred to have resulted in the situation observed. Figure 2c can be explicitly mentioned, so that readers know that this gives an explanation. The term “triangular translocation” is still used, but it is non-standard and does not convey a clear meaning. The production of a ring hexavalent is clear, and it is unhelpful to use words that make the meaning less clear.

We inserted one diagram showing three translocations. Thanks for this suggestion.

“Triangular” was changed to “three or three sequential” translocations as stated above.

3.2. A hexavalent ring at male meiosis 165 To investigate the number of translocations, we observed paring figures of the first 166 meiotic chromosomes in male testes. It was found that the meiotic karyotype comprised 167 one large ring-shaped hexavalent, together with ten ring bivalents at the first meiotic metaphase (Fig. 1c and Fig.S1b, c). Three chromosome pairs must therefore be involved in the translocations. In addition, we detected telomeres in all the chromosomes involved in the 171 translocations (Fig.S1c, d). Therefore, the most likely, expected order of translocations to 172 form the hexavalent ring is as follows: the large part of the long arm of chromosome 1, 173 including the terminal tip, was translocated to the long arm of chromosome 7, in which 174

5 of 10

the terminal tip (including two bands) of long arm was broken before the fusion and 175 moved to the short arm of chromosome 3, 1one band of whose short arm terminus was broken before the fusion and moved to the long arm of 2chromosome 1, which lost the large part of the long arm (Fig. 1d). In the remaining parts of the chromo- 178 some 1 long arm and of the chromosome 3 short arm, one paracentric inversion also occurred 179 (Figs. 1d and S1a).

1 Broken was the terminal tip (region) including a band, but not “a band”.

2 “chromosome, which lost” can be interpreted as “chromosome, and then it lost”. These should be connected by “that” without comma.

“missed” is better than “lost” because “lost” means “disappeared from the world and the destination is unknown”.

181 3.3. Direct proof of the chromosome members involved in triangular translocations 182 To directly identify the chromosome members involved in the triangular transloca- 183 tions, we performed hybridization painting using the microdissected chromosomal 184 DNA probes. Both of the chromosome 1 and 3 probes (green and red, respectively) 185 painted parts of the hexavalent (Fig. 2a-1).

COMMENT: Here, the next sentences appear to be garbled. I attempted to guess the meaning, but the authors need to check that it is correct. The chromosome 1 probe painted chromosome pair 1 in the female (Fig.2a-2), and also the heteromorphic chromosome pair 1 and the long arm of the longer chromosome 7 in male (green arrow in Fig.2a-3). The chromosome 3 probe painted chromosome pair 3 in females , as expected(Fig.2a-2), and also the heteromorphic chromosome pair 3 and the short arm of the shorter chromosome 1 in males (pink arrow in Fig. 2a-3). These results directly prove that chromosomes 1 and 3 were involved in the male specific translocations from chromosome 1 to 7 and from chromosome 3 to 1. Next, hybridization using the hexavalent DNA probe painted chromosome pairs 1, 3 193 and 7 in female, while heteromorphic chromosome pairs 1, 3 and 7 in male (Fig.2a-4, 5, 194 6). Thus, it is concluded that the male-specific triangular translocations among chromo- 195 somes 1, 3 and 7 created a system of six sex-chromosomes, ♂X1Y1X2Y2X3Y3 - 196 ♀X1X1X2X2X3X3 (Fig.2b and c). 197 198199 Figure 2c. 

Thank you for this correction. We exchanged “while” with “and”in L188 190 and 195, and added “in male” in L191.

3.4. No male DNA specialization on the three Y chromosomes 216 To examine the extent of chromosomal specialization of the three Y chromosomes, 217 we observed C-banded karyotypes. No Y-specific heterochromatin was detected 218 on the three Y chromosomes (Fig. S2). In addition, complementary genomic hybridization (CGH) on the male and female chromosomes detected neither male nor female specific signals (Fig. S3). These results show that the three Y chromosomes have not yet accumulated male-specific DNA or heterochromatin.

We removed “stained” and changed “had” to “have” in L219.

3.5. Population variation in the sex-chromosome system 224 To elucidate the sex-chromosomal variations within this species, we observed the karyotypes of the frogs collected from two other populations (Table S1), of which one 226 was located just south of the first population (New Taipei), and the other was around the central region of the island (Nantou Ren Ai). We observed no translocations or heteromorphic sex-chromosomes in the males or females based on the late replication and C-banding patterns (Figs S4–6). This result suggests that the translocations in males are restricted in some geographic populations of the species and originated very recently.

We changed to “originated very recently” in L232-233.